# Intercalant-independent transition temperature in superconducting black phosphorus

R. Zhang[1], J. Waters[2], A.K. Geim[1] & I.V. Grigorieva[1]

Research on black phosphorus has been experiencing a renaissance over the last years, after the demonstration that few-layer crystals exhibit high carrier mobility and a thickness-dependent bandgap. Black phosphorus is also known to be a superconductor under high pressure exceeding 10 GPa. The superconductivity is due to a structural transformation into another allotrope and accompanied by a semiconductor-metal transition. No superconductivity could be achieved for black phosphorus in its normal orthorhombic form, despite several reported attempts. Here we describe its intercalation by several alkali metals (Li, K, Rb and Cs) and alkali-earth Ca. All the intercalated compounds are found to be superconducting, exhibiting the same (within experimental accuracy) critical temperature of $3.8 \pm 0.1$ K and practically identical characteristics in the superconducting state. Such universal superconductivity, independent of the chemical composition, is highly unusual. We attribute it to intrinsic superconductivity of heavily doped individual phosphorene layers, while the intercalated layers of metal atoms play mostly a role of charge reservoirs.

[1] School of Physics and Astronomy, University of Manchester, Oxford Road, Manchester M13 9PL, UK. [2] School of Earth, Atmospheric and Environmental Sciences, University of Manchester, Oxford Road, Manchester M13 9PL, UK. Correspondence and requests for materials should be addressed to I.V.G. (email: irina.grigorieva@manchester.ac.uk).

Bulk black phosphorus (BP) is the most thermodynamically stable phosphorus allotrope[1–3] with a moderate direct bandgap (0.3 eV for bulk BP[2], increasing to 2 eV for monolayer phosphorene[4]). Recent demonstrations of high mobilities[5,6], quantum oscillations[7], thickness-dependent gap[6] and field-effect transistors with high on-off ratios[8] using few-layer BP has led to a strong wave of interest in this material in its quasi-two-dimensional form. Furthermore, the bandgap of few-layer BP has been predicted to be tuneable by strain[9] and electric field[10], whereas surface doping of few-layer BP with potassium was found to result in a metallic state[11]. While the interest in semiconducting BP is focused on its promise for nanoelectronics and nanophotonics[3], metallic BP was predicted to have sufficient electron–phonon coupling to become a superconductor[12,13].

Metallic state and superconductivity in phosphorous were previously achieved in the 1960s (refs 14,15) by applying high pressure. The transition to the metallic state was shown to be associated with the structural transition from the orthorhombic to simple-cubic crystal lattice, which actually means that the latter material was no longer BP but somewhat closer to white phosphorous that also has a cubic lattice[16]. The superconducting transition was found at 4.8 K at 10 GPa and the transition temperature, $T_c$, could be further increased to 9.5 K at 30 GPa (refs 14,15). The question whether it is possible to induce superconductivity in BP by other means has remained open. Recent experiments on electrostatically doped thin flakes of BP (ref. 17) did not find superconductivity, despite being able to achieve carrier concentrations well above $10^{13}$ cm$^{-2}$, sufficient to induce superconductivity in, for example, electrostatically doped $MoS_2$ (ref. 18). There were also several attempts to obtain intercalated compounds of BP by reacting it with alkali metals[19] (K, Cs and Li), iodine[20], and $AsF_5$ (ref. 21) and by electrochemical lithiation[22]. No stable intercalation compounds could be achieved in either of these studies, and no superconducting response was reported or discussed.

To overcome the problem of disintegration of BP crystals when in contact with highly reactive vapours of alkali metals, we have employed an alternative technique of liquid ammonia intercalation[23,24]. This allowed us to achieve successful intercalation of BP crystals with several alkali and alkali-earth metals: lithium (Li), potassium (K), rubidium (Rb), caesium (Cs) and calcium (Ca). All our intercalation compounds show superconductivity with $T_c$ of 3.8 K, the same value within our

experimental accuracy of ± 0.1 K. We emphasize that $T_c$ does not depend on the intercalating metal, which indicates that the superconductivity is an intrinsic property of electron-doped phosphorene (individual layers of BP), as described below.

## Results

**Sample preparation and characterization.** The atomic arrangements expected for pristine and intercalated BP are shown schematically in Fig. 1a,b (refs 13,25–27). BP consists of weakly bonded phosphorene layers, within which covalently bonded P atoms form a honeycomb network, similar to graphene, but each layer is puckered with a zigzag-shaped edge along the $x$ axis and an armchair-shaped edge along the $y$ axis[27]. To achieve metal intercalation and obtain intercalated compounds $M_xP$ (here M stands for Li, K, Rb, Cs and Ca), crystals of BP were immersed in a metal–liquid ammonia solution at a temperature of −78 °C in a dry-ice/isopropanol bath (see Methods for details). The starting solution has a characteristic deep-blue colour due to dissociation of metal atoms into solvated cations ($M^+$) and solvated electrons ($e^-$), (ref. 28). As intercalation proceeds, the solution gradually loses its colour, which is known to be an indication of metal ions moving into the van der Waals gaps between the layers of host crystals[23]. This allowed us to monitor the process visually. Intercalation was also apparent from significant swelling of the crystals, with an expansion along the $z$ axis (for example, a sample with thickness of 0.4 mm expanded to 1.0 mm). The intercalation process started at the surface of the host crystal and, as described below (see also ref. 23), resulted in a superconducting fraction below 100% (up to 10% if thin BP crystals were used). We estimate that only ~10 μm thick surface layers were fully intercalated.

To determine the chemical composition of our intercalated compounds, we used energy dispersive X-ray spectroscopy (EDS). EDS typically probes a few-micron thick surface layer and is therefore the most appropriate method to determine the stoichiometry of our surface-intercalated samples. EDS spectra for our $M_xP$ are shown in Fig. 1e and the corresponding elemental maps in Supplementary Fig. 1. The distribution of intercalated metal atoms is fairly uniform for all $M_xP$ compounds, and our EDS analysis yielded the average concentrations of 20, 17, 18 and 8 at.% for K, Rb, Cs and Ca intercalated crystals, respectively (typical error in these measurements was 3 at.%).

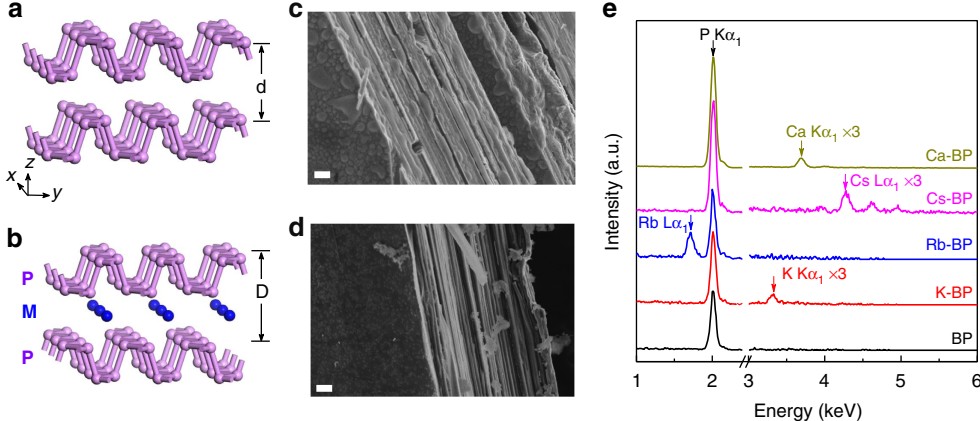

**Figure 1 | Structure and composition of pristine and intercalated BP.** (**a,b**) Schematic structure of BP before (**a**) and after (**b**) intercalation. 'P' stands for phosphorus atoms and 'M' represents intercalating metals Li, K, Rb, Cs or Ca. The shown positions of metal atoms correspond to the results of first-principle calculations[13,25,26]. Those suggested that the adjacent phosphorene layers should slide with respect to each other, effectively changing the stacking order of phosphorene layers and providing a maximum space for the metal ions. (**c,d**) Cross-sectional scanning electron microscopy images of pristine (**c**) and Cs-intercalated (**d**) BP. Scale bars, 1 μm. (**e**) Typical EDS spectra for pristine and intercalated BP.

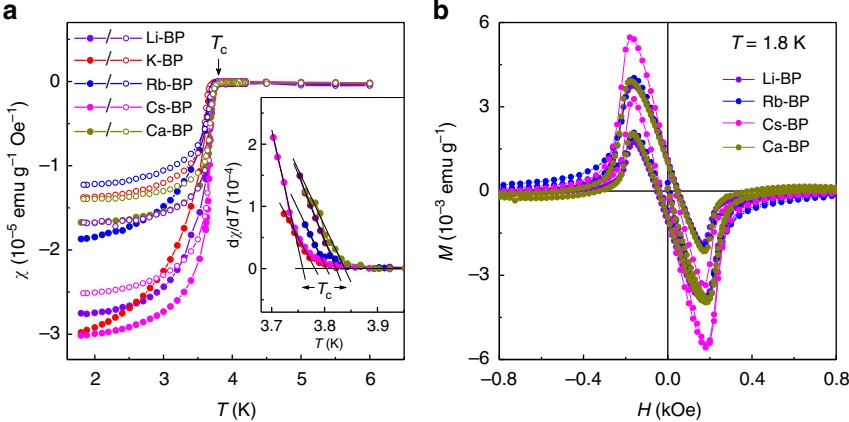

**Figure 2 | Superconductivity in intercalated BP. (a)** Temperature dependence of magnetic susceptibility $\chi$ for Li, K, Rb, Cs and Ca intercalation. Shown are ZFC (solid symbols) and FC (open symbols) measurements for an in-plane magnetic field of 10 Oe. The arrow indicates $T_c$ as determined from $d\chi/dT$. The inset shows how $T_c$ was determined for individual samples: It is defined as the sharp change in $d\chi/dT$ (see also Supplementary Fig. 4). **(b)** Magnetization $M$ as a function of magnetic field for our intercalated compounds.

Within this accuracy, the found concentrations correspond to the same composition of monovalent alkali-metal intercalated BP (namely, $K_xP$, $Rb_xP$ and $Cs_xP$, where $x \approx 0.2$) and an approximately twice lower content for divalent Ca ($Ca_xP$ with $x \approx 0.1$). It was not possible to determine the concentration of Li using EDS because the technique is insensitive to elements with atomic numbers $< 5$. The above compositions are stoichiometric (or stage I, ref. 29), in agreement with the first-principles calculations for Li, Na and Mg intercalation[13,25,26], which predicted that the metal atoms should occupy certain positions between the puckered hexagonal layers, as shown in Fig. 1b.

We have also used X-ray diffraction (XRD) to probe the interlayer spacing after intercalation. For pristine BP, our XRD spectra exhibited three main characteristic peaks at $2\theta = 17.1°$, $34.4°$ and $52.5°$, corresponding to (020), (040) and (060) crystallographic planes, respectively (Supplementary Fig. 2a). This gives a layer spacing of 5.18 Å, in agreement with other studies[3,27]. However, after intercalation, while the peaks characteristic of pristine BP have almost disappeared, as expected, no new peaks corresponding to an expanded crystal lattice are visible. We attribute this to the fact that intercalation is limited to $\sim 10\,\mu m$ thick layers near the surface, while X rays probe the entire volume of the crystals. Let us mention that such disappearance of XRD peaks is not unusual and was also reported for BP electrochemically intercalated with $Na^+$ (ref. 30). However, in contrast to the latter study, in our experiments the peaks corresponding to pristine BP re-appeared after exposure of the intercalated samples to air for about 100 h (Supplementary Fig. 2b) and all signs of superconductivity disappeared (Supplementary Fig. 3). This indicates that, unlike the irreversible electrochemical intercalation[22,30], our procedure preserved the original structure of the puckered BP layers.

**Magnetization measurements**. To detect the superconducting response, we used SQUID magnetometry (Methods section). Figure 2a shows the temperature dependence of dc magnetic susceptibility $\chi = M/H$ for all our intercalated BP compounds under zero-field cooling (ZFC) and field-cooling (FC) conditions, and Fig. 2b provides examples of the magnetization as a function of applied magnetic field $H$. Here $M$ is the magnetic moment. Both ZFC and FC curves show a sharp increase in diamagnetic susceptibility at 3.8 K, characteristic of a superconducting transition. To find the transition temperature, we used the derivatives of the ZFC curves, and $T_c$ for each intercalated sample

was determined as the temperature at which $d\chi/dT$ exhibited a sharp increase (see inset in Fig. 2a and Supplementary Fig. 4a). This yielded $T_c = 3.8 \pm 0.1\,K$ for all $M_xP$, independent of the metal and whether it is monovalent or divalent. Note that we found no correlations between the observed slight variations in $T_c$ and the intercalant (the same spread of $\pm 0.1\,K$ in $T_c$ was found for different samples intercalated with the same metal; Supplementary Fig. 4b).

To further characterize the superconductivity in our samples, we measured ac susceptibility, $\chi_{ac}$, as a function of both $H$ and $T$. This allowed us to accurately determine the orientation-dependent critical field, $H_c$, corresponding to the disappearance of superconductivity, as shown in Fig. 3a, where the dc magnetization and ac susceptibility for a Cs-intercalated BP are shown together (more examples of ac susceptibility are shown in Supplementary Fig. 5). These measurements were also used to find the temperature and orientation dependence of $H_c$ as shown in Fig. 4. The $H_c$ values were determined both from $\chi_{ac}(H,T)$ (Fig. 3 and Supplementary Fig. 5) and $M(T,H)$ curves (Fig. 4a and Supplementary Fig. 6), with good agreement between the values obtained by the two methods. Within our experimental accuracy, both parallel and perpendicular $H_c(T)$ are universal for all $M_xP$, that is, do not depend on the intercalant. This is similar to their intercalant-independent $T_c$.

Let us note that the superconducting fraction in the measurements shown in Figs 2–4 was rather small, $\sim 1$–2%, which we attribute to the dynamics of the intercalation process, as already mentioned above. It is much easier for metal ions to start intercalating between outer phosphorene layers, so that the process proceeds from the surface of each crystal into its bulk. The data in Figs 2–4 were obtained on relatively large individual crystals, $\sim 3 \times 2 \times 0.3\,mm$, in order it would be possible to determine their orientation with respect to the applied magnetic field. By intercalating thinner crystals, with larger surface-to-volume ratios (Methods section), we were able to increase the superconducting fraction to $\sim 10$% (Supplementary Fig. 7), which confirms the bulk nature of superconductivity.

## Discussion

There are several notable features in the $H$ dependence of intercalated BP's magnetization, both dc and ac (Figs 2b and 3). First, the $M(H)$ curves are practically identical, apart from somewhat different absolute values of the maximum diamagnetic moment for different intercalating metals in Fig. 2b, which is

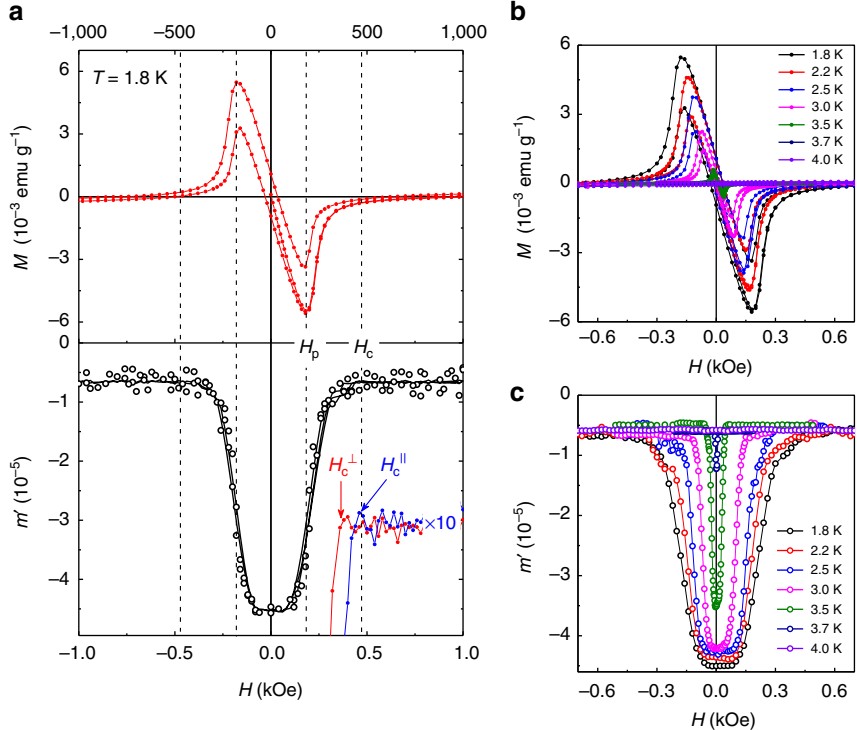

**Figure 3 | Magnetic field dependence of magnetization in $M_xP$.** (**a**) dc magnetization, $M$, and the real part of ac susceptibility, $m'$, as a function of in-plane magnetic field, $H$. For clarity, only data for Cs-intercalated BP are shown. Open symbols are data points measured as the dc field varied from $-1$ to $+1$ kOe and from $+1$ to $-1$ kOe. Solid lines show the same data smoothed using Savitzky–Golay filter. $H_p$ and $H_c$ denote the penetration and critical magnetic field, respectively. Inset: Zoomed-up $m'$ near the critical field in two orientations, with the arrows indicating $H_c^{\parallel}$ and $H_c^{\perp}$. (**b,c**) Temperature dependences of $M(H)$ and $m'(H)$. Only data for $Cs_xBP$ are shown. Both types of measurements give the same values of $T$-dependent $H_p$ and $H_c$.

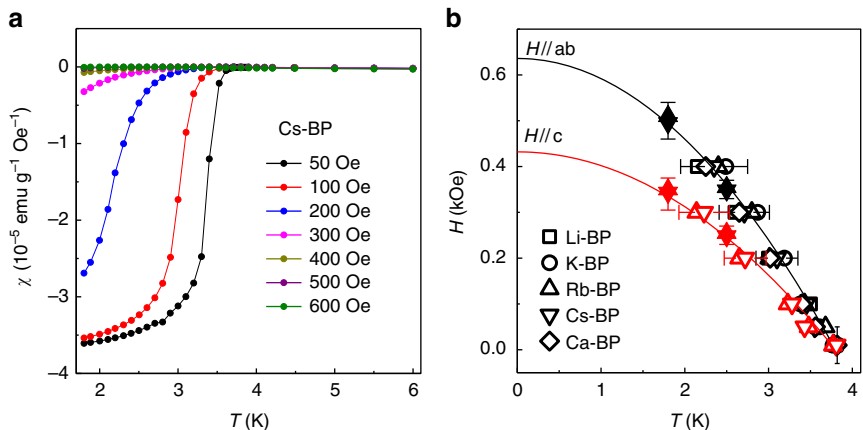

**Figure 4 | Transition temperature as a function of magnetic field.** (**a**) Temperature dependence of ZFC dc susceptibility $\chi$ at different $H$ applied parallel to the surface (ab plane). For clarity, only data for Cs-intercalated BP are shown. (**b**) Parallel and perpendicular critical fields as a function of temperature for all $M_xP$ compounds in this study. Error bars indicate the accuracy of determination of the critical temperature from $\chi(T)$ curves such as those shown in **a** (open symbols) and of the critical field from $m'(H)$ (solid symbols). Solid curves are fits to the BCS expression $H_c \propto 1 - (T/T_c)^2$.

related to different superconducting fractions. Indeed, all the compounds exhibit the same penetration field, $H_p$ (1.8 K) $\sim 180$ Oe (determined from dc magnetization curves as shown in Fig. 3a), and the same magnetic field corresponding to the disappearance of the diamagnetic response, $H_c$ (1.8 K) $\sim 500$ Oe. Second, the sharp fall in $M$ just above $H_p$ is reminiscent of the behaviour of type I, rather than type II, superconductors. In the former case, the magnetic field penetrates either uniformly at the thermodynamic critical field, $H_m$, destroying superconductivity or, for a finite demagnetization factor, in the form of normal domains[31]. The relatively rapid decrease is also seen in

ac susceptibility (Fig. 3). However, there is a clear difference from a typical type I superconductor with the Ginzburg–Landau parameter $\kappa \ll 1$ if we compare our $M(H)$ curves in Fig. 2 with $M(H)$ for indium that we used as a reference (Supplementary Fig. 8). The 'tail' on the $M$ and $m'$ curves seen at $H > 300$ Oe corresponds to small but still finite diamagnetic response above $H_p$. This behaviour suggests that the intercalated BP is probably a borderline type I/type II superconductor with $\kappa$ close to the critical value of $\approx 0.7$, similar to Nb (ref. 32). Third, and very unusually, the $M(H)$ curves for all $M_xP$ compounds display practically the same hysteresis (trapping of the magnetic flux).

The latter corresponds to pinning of the magnetic flux by defects or impurities[31] and, therefore, is sensitive not only to a particular chemical compound, but also to structural details of individual samples. The fact that, in our case, different intercalated compounds trap the same amount of flux at $H = 0$ also points in the direction of type I superconductivity because pinning of superconducting domains is known to be less sensitive to crystal imperfections[31,33].

To relate the appearance of superconductivity with the structural changes in BP, which occur as a result of intercalation, we used Raman spectroscopy (Supplementary Fig. 9). The well-known signatures of phonon modes in BP (refs 30,34) could be compared before and after intercalation. Raman spectroscopy is sensitive to the properties of a few near-surface layers and, therefore, probes the same part of the crystals as EDS and magnetization measurements. Atomic displacements corresponding to the Raman modes are illustrated in Supplementary Fig. 9a. There are three main phonon modes that in our pristine BP result in peaks at 362, 440 and 467 cm$^{-1}$ (see Supplementary Fig. 9b). From comparison of the Raman spectra for pristine and intercalated BP it is clear that intercalation has only a small effect on atomic vibrations of P atoms. The modes are slightly broadened but essentially unchanged, which is consistent with the preserved structure of phosphorene layers and in agreement with the mentioned experiments in which samples exposed to air recovered their initial structural state. At the same time, one can see a clear shift of all three modes to lower frequencies, corresponding to phonon softening within phosphorene layers. Such softening is expected as a result of the doping of phosphorene, which according to the first-principle calculations[13,25] should result in charge transfer of 0.8e$^{-}$ per P atom for monovalent and 1.5e$^{-}$ for divalent donors.

The most surprising result of our work is the practically identical superconducting characteristics of all the tested intercalated compounds, independent of the mass and atomic radius of the intercalating metal and its valency, that is, the lack of any isotope-like effect. This is in stark contrast to other superconducting intercalated compounds, of which intercalated graphite is an archetypal example and has obvious structural similarities with BP. In the case of intercalated graphite, its superconducting properties are strongly dependent on the chemical composition, with the critical temperature ranging from 0.02 K for Cs-intercalated graphite to 11.5 K for CaC$_6$ (ref. 35) and LiC$_6$ not superconducting at all. The mechanism of superconductivity in the intercalated graphite compounds is well understood[36–38]. In terms of the standard phonon-mediated superconductivity, superconductivity in graphite cannot be achieved within individual electron-doped graphene layers because the frequencies of in-plane graphene phonons are too high to lead to efficient electron–phonon coupling, whereas the planar structure of graphene forbids coupling between the π* electronic states and softer out-of-plane vibrations[36]. Therefore, an incomplete ionization and a partially occupied metal-derived electronic band is a key for achieving superconductivity in intercalated graphite[36,37], and the superconductivity cannot be attributed to either the graphene π* band or the metal-derived electronic band alone. Instead, the interaction between these two bands plays a critical role, according to theory, and the transition temperature is strongly dopant dependent.

The situation in BP is quite different. The experiment clearly shows that the chemical composition of intercalated BP is irrelevant for the occurrence of superconductivity, and one has to find a physical mechanism relying on heavy electron doping of individual phosphorene layers. This is in good agreement with theoretical expectations. Indeed, the puckered structure and $sp^3$ hybridization of phosphorene layers remove the symmetry

constraints on electron–phonon coupling[13], with more phonon modes able to contribute, increasing the overall coupling constant, $\lambda$. In addition, electron transfer from intercalant atoms to phosphorene softens its vibrational modes, especially the sublayer breathing modes, further enhancing $\lambda$ without the need of changing the crystal symmetry of phosphorene layers[12,13]. In our experiments such phonon softening is clear from the Raman spectra, where a similar frequency shift is observed for all intercalants (Supplementary Fig. 9b). Furthermore, recent calculations for Li-intercalated bilayer phosphorene[13] indicate that the electronic bands near the Fermi level of intercalated BP are mainly π*-like bands derived from phosphorene, while the contribution to the density of states from the metal-derived band is very small. This basically means that the superconductivity is intrinsic to individual phosphorene layers rather than the entire compound, in agreement with our experiments. Let us mention that the theoretical explanation is also consistent with recent angle-resolved photoemission spectroscopy (ARPES) measurements of the BP intercalated with K (ref. 11) or Li (ref. 39), which found no additional interlayer bands that could be attributed to the alkali metals.

The intercalant-independent $T_c$ implies the same doping level induced within phosphorene layers. The situation is somewhat similar to equal doping and intercalant-independent $T_c$ in K, Rb and Cs-intercalated MoS$_2$ (refs 40,41). This is also in agreement with theory. The first-principle calculations[25] found an optimum charge transfer for monovalent alkali atoms (Li and Na) of 0.8e$^{-}$ per P atom and for divalent Mg of 1.5 e$^{-}$. It is reasonable to expect the same electron doping for K, Rb and Cs that were not covered in the calculations and a close doping for divalent Ca that is chemically similar to Mg and intercalates in twice smaller concentrations than alkali metals. Furthermore, several studies, both theoretical[25] and experimental[22] found that the stoichiometric composition M$_{0.2}$P found in our studies is the maximum—and probably optimum—alkali content, for which the structure of phosphorene layers is preserved and the intercalation can be reversed. If more metal atoms are forced into the space between phosphorene layers by, for example, electrochemical intercalation[22,30], this leads to irreversible changes including chemical bonding and formation of new alloys. Taken together, the available evidence suggests that the amount of intercalating metal atoms in BP is determined primarily by an electrostatic-like equilibrium between the charged phosphorene layers and metal ions. In our case, this conclusion is supported by the factor-of-2 difference in the concentration of monovalent and divalent intercalants as found experimentally.

In summary, we have achieved successful intercalation of BP with several alkali and alkali-earth metals using the liquid ammonia method. The intercalated compounds exhibit universal superconducting properties: same $T_c = 3.8 \pm 0.1$ K, same critical fields $H_c^{\parallel}(0) \sim 630$ Oe and $H_c^{\perp}(0) \sim 440$ Oe and even similar flux pinning. The lack of variations strongly suggests a physical mechanism behind the superconductivity, such as doping of phosphorene layers by intercalating atoms. In this respect, intercalated BP is very different from extensively studied intercalated graphite compounds, despite their structural similarities. In the latter case, both electronic and phonon contributions from intercalated metal layers play a key role in inducing superconductivity. In intercalated BP, the superconductivity can entirely be attributed to electron-doped phosphorene, whereas changes in the electron and phonon spectra induced by intercalation play little role, according to the experiment and in qualitative agreement with theory. Nevertheless, the degree to which the superconducting behaviour disregards the chemical composition is puzzling and perhaps suggests some

more fundamental rules at play behind the observed superconductivity.

## Methods

**Alkali-metal intercalation.** To achieve intercalation, BP crystals (99.998%, Smart Elements) and a desired metal (99.95% Li, K, Rb, Cs and Ca from Aldrich) were sealed in a quartz tube under the inert atmosphere of a glovebox with oxygen and moisture levels less than 0.5 ppm. The reactor tube was then evacuated to $\approx 10^{-5}$ mbar, connected to a cylinder of pressurized ammonia (99.98% from CK Gas) and placed in a bath of dry-ice/ethanol ($-78\,°C$). As gaseous ammonia was allowed into the reactor tube, it condensed onto the reactants (BP and the metal), dissolving the metal and forming a deep-blue solution. The colour depth was used as an indicator of the concentration of dissolved metal[28]. To prevent contamination with oxygen or moisture, the empty space in the reactor tube left after ammonia condensation was filled with argon gas (Zero-Grade from BOC). The tube was kept in the dry-ice/ethanol bath for 48 h, after which the system was warmed up, ammonia and argon evacuated and the intercalated BP recovered inside the glovebox. As alkali-metal intercalated compounds are extremely sensitive to oxygen and moisture, they were handled in the inert atmosphere of the glovebox, immersed in paraffin oil or protected by using sealed containers. As a reference, we tested our method by intercalating $MoS_2$ with different alkalis, which produced superconducting samples with $T_c \approx 6.9\,K$, in agreement with literature[41]. In addition, to ensure that the observed superconductivity could not be related to accidental contamination, we used mixtures of BP and alkali metals (without placing them in liquid ammonia) and measured their magnetization. No superconducting response could be detected unless the intercalation reaction was induced.

We note that in metal-ammonia reactions there is a very small but finite chance that ammonia can decompose with a possible formation of minute quantities of metal hydrides. In the case of intercalated graphite, one of the hydrides ($C_8KH$) was found to exhibit superconductivity with $T_c \approx 0.2\,K$ (ref. 42). In the case of BP, even if there were an analogous contaminant, the found $T_c$ in our $A_xBP$ is a factor of 20 higher, so that this kind of possible contamination would not account for the observed superconductivity. As for a possible phosphorus hydride ($PH_3$), this only becomes superconducting at very high pressures, $>200\,GPa$ (ref. 43), and is a semiconductor in ambient conditions.

**Chemical and structural characterization.** The chemical composition was determined using EDS (Oxford Instruments X-Max detector integrated with Zeiss' Ultra scanning electron microscope). The metal content was determined at 10 or more randomly selected areas on each sample and the average was taken as the intercalant concentration.

XRD measurements were performed using Bruker D8 Discover diffractometer with Cu K$\alpha$ radiation ($\lambda = 1.5406\,Å$). Before each measurement, samples were roughly ground and mixed with paraffin oil, and then sealed in an airtight XRD sample holder (Bruker, A100B36/B37) inside the glovebox. XRD spectra were obtained at room temperature in the $2\theta$ range of $5$–$75°$ with a step of $0.03°$ and a dwell time of 0.5 s per step.

Raman spectroscopy was performed at room temperature using the Renishaw inVia Reflex system with a 532 nm laser. All spectra were recorded using a power of $\sim 1\,mW$ with the samples sealed between two thin glass plates.

**Magnetization measurements.** Magnetization measurements were carried out using Quantum Design MPMS-XL7 SQUID magnetometer. To protect the intercalated samples from degradation, they were immersed in paraffin oil and sealed inside polycarbonate capsules in a dry argon atmosphere of the glovebox. Measurements were carried out in a dc magnetic field applied either parallel or perpendicular to the crystal surface. Temperature dependences were measured in ZFC and FC modes. In the former, the samples were cooled in zero magnetic field from $\sim 10$ to $1.8\,K$, then the field $H$ was applied and $M$ was measured as the temperature increased from 1.8 to, typically, $6\,K$. In latter mode, $M(T)$ was measured as the temperature decreased from above 6 to $1.8\,K$. The ac susceptibility was measured using ac magnetic fields (typically, 1 Oe at 8 Hz) applied parallel to the dc field.

**Superconducting fraction versus sample dimensions.** To estimate the superconducting fraction in different samples, we used the initial (Meissner) slope of the $M(H)$ curves (for a 100% superconducting sample the slope should be $\sim 1/4\pi$). This yielded a fraction of $\sim 1$–2% if relatively large ($\sim 3 \times 2 \times 0.3\,mm$) crystals were intercalated. To increase the intercalated volume, we used two different methods. First, a crystal of BP was ground into a powder consisting of many smaller platelet-shaped crystals that were then intercalated as described above. This increased the superconducting fraction by a factor of $\sim 3$. However, we believe that the smallest BP crystals within this powder were non-superconducting, reacting with minute amounts of oxygen and moisture left even under inert atmosphere[5]. The second and more successful approach was to break up BP crystals into smaller ones during the intercalation process. To this end, the liquid

ammonia solution was subjected to mild shaking using a magnetic stirrer. This allowed an increase in the superconducting fraction to $\sim 10\%$.

**Data availability.** The data that support the findings of this study are available from the corresponding author on request.

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

## Acknowledgements

We acknowledge support from the UK Engineering and Physical Sciences Research Council (EPSRC)—EP/N010345/1; EC-FET Graphene Flagship, Grant Agreement No. 604391 and Lloyd's Register Foundation. R.Z. acknowledges support from the China Scholarship Council. We thank Kai-Ge Zhou for his help with the artwork.

## Author contributions

R.Z. and I.V.G. conceived the project. R.Z. developed the intercalation process, prepared and characterized all samples and carried out magnetization measurements. J.W. carried out X-ray characterization. I.V.G. supervised the work. R.Z., I.V.G. and A.K.G. analysed the data and wrote the paper.

## Additional information

**Competing interests:** The authors declare no competing financial interests.

**How to cite this article**: Zhang, R. *et al.* Intercalant-independent transition temperature in superconducting black phosphorus. *Nat. Commun.* **8,** 15036 doi: 10.1038/ncomms15036 (2017).

**Publisher's note**: 

