## [Peer Review File · Nature Communications]

Reviewers' Comments:

Reviewer #1 (Remarks to the Author)

This paper reports the discovery of a new superconductor and the application of a metal-ammonia method of intercalation used in transition dicalcogenides and graphite. This technique is still in its infancy in this particular matrix. The authors make this clear and have clearly demonstrated within the paper. They have applied a suite of analytical methods to explore the structural and electronic states. Clearly there is more work to be done on these aspects, however this is a discovery paper and the authors work provides an important superstructure for future work.

The principal conclusion is intercalated black phosphorus is a superconductor and this is certainly demonstrated through the magnetometry. Type I or type II superconductor is still not absolutely clear from the data. When the sample growth is refined this will become clearer. Perhaps more significantly is the identification of the principal role played by the phosphorus in the superconducting ground state. More explicitly that the role played by the intercalant; a role different from that identified in graphite intercalation compounds which is an active role. The Raman studies do not reveal any significant softening of the three main phonon modes of the phosphorus sheets. Again, at this stage of discovery, this presents technical challenges to search for modes closer to the gamma point and will be needed in future work. At this stage it looks like the main role of the intercalant is electron doping, and the absence of a mass dependency of T_c based on intercalant points strongly to a superconducting ground state based purely on phosphorus.

Are the results of this work "novel"? Yes they certainly are new and important.

This is important to the scientists in the field of the superconducting ground state since there is a fundamental question concerning the phonon modes important to the superconducting coupling of electron states. The authors say this very clearly in the closing two sentences of this paper.

In short, I strongly support the publication of this paper. There are one or two minor typos but these I am sure will have been picked up.

Reviewer #2 (Remarks to the Author)

Reviewer #2

In this manuscript, the authors present a superconducting behavior in the intercalated BP with different alkali metals (Li, K, Rb, Cs) and alkali-earth Ca. They find that all doped samples are superconducting and exhibiting exactly the same critical temperature of 3.8 ± 0.1 K. The authors attribute the superconductivity found in the doped sample to the role of intercalated layers of metal atoms. If these experimental results and the corresponding analysis are confirmed to be correct, this would be an interesting finding. However, this manuscript totally failed to convince me for what they described. Therefore, I don't recommend this manuscript to be published in NC. Some reasons are listed below:

1. I don't think that the title of "Observation of doping-induced superconductivity in phosphorene" is appropriate to describe the intercalated BP system studied. This can be misleading, because the achieved doped surface (in micron scale) of the black P in this study should be quite different from a doped phosphorene (in nano scale).
2. If the T_c is exactly 3.8 K for all samples, then that should be impossible if they are really doing what they say that they are doing - intercalating the alkali ions. Given that the T_c s are all the same, it seems like that it would be fair for them to have to prove that they really did intercalate the black P to make the superconductor.
3. It can't be excluded that the unchanged T_c are originated from the same thing rather than the doped phosphorene as they expected, for instance, the contamination.

Reply to comments of Reviewer #1

Reviewer #1 (Remarks to the Author):

This paper reports the discovery of a new superconductor and the application of a metal-ammonia method of intercalation used in transition dicalcogenides and graphite. This technique is still in its infancy in this particular matrix. The authors make this clear and have clearly demonstrated within the paper. They have applied a suite of analytical methods to explore the structural and electronic states. Clearly there is more work to be done on these aspects, however this is a discovery paper and the authors work provides an important superstructure for future work.

The principal conclusion is intercalated black phosphorus is a superconductor and this is certainly demonstrated through the magnetometry. Type I or type II superconductor is still not absolutely clear from the data. When the sample growth is refined this will become clearer. Perhaps more significantly is the identification of the principal role played by the phosphorus in the superconducting ground state. More explicitly that the role played by the intercalant; a role different from that identified in graphite intercalation compounds which is an active role. The Raman studies do not reveal any significant softening of the three main phonon modes of the phosphorus sheets. Again, at this stage of discovery, this presents technical challenges to search for modes closer to the gamma point and will be needed in future work. At this stage it looks like the main role of the intercalant is electron doping, and the absence of a mass dependency of T_c based on intercalant points strongly to a superconducting ground state based purely on phosphorus.

Are the results of this work “novel”? Yes they certainly are new and important.

This is important to the scientists in the field of the superconducting ground state since there is a fundamental question concerning the phonon modes important to the superconducting coupling of electron states. The authors say this very clearly in the closing two sentences of this paper.

In short, I strongly support the publication of this paper. There are one or two minor typos but these I am sure will have been picked up.

We would like to thank the referee for this kind assessment.

Reply to comments of Reviewer #2

In this manuscript, the authors present a superconducting behavior in the intercalated BP with different alkali metals (Li, K, Rb, Cs) and alkali-earth Ca. They find that all doped samples are superconducting and exhibiting exactly the same critical temperature of 3.8 ± 0.1 K. The authors attribute the superconductivity found in the doped sample to the role of intercalated layers of metal atoms. If these experimental results and the corresponding analysis are confirmed to be correct, this would be an interesting finding. However, this manuscript totally failed to convince me for what they described. Therefore, I don't recommend this manuscript to be published in NC.

We thank the referee for reading our manuscript and for their comments. We are truly disappointed that we failed to convince him/her in the validity of our results. However, we are not sure what else could be done, even in principle.

First of all, our experiments are 100%(!) reproducible, that is, we observed the same superconducting transition on numerous samples in many repetitive experiments. At this stage of our experience with intercalation, there are no samples that fail to exhibit superconductivity. Of course, we are happy to provide our samples for independent assessment to the referee or other interested parties.

Second, let us mention that we have had extensive experience (4 years) in studying intercalated layered compounds. Some of our experiments were reported in *Nano Lett.* **16**, 629 (2015) and *Sci. Reports* **6**, 23254 (2016). As a reference, we also made many samples of graphite and MoS_2 , which were

intercalated with alkali and alkali-earth elements and found the same T_c 's as reported previously in the literature. Following the referee's comment, we added the info about our reference experiments to the revised manuscript (in Methods on p. 9).

Third, we need to point out that our results (intercalant-independent T_c) are consistent with those for intercalated MoS_2 , also a heavily doped semiconductor. The latter shows superconductivity with the same critical temperature, $T_c \approx 6.9\text{K}$ when intercalated with K, Rb and Cs. This result was reproduced in several reports [J. Woolam, R. Somoano, Mater. Sci. Eng. **31**, 289 (1977) and references therein] and interpreted as a rigid shift of the electronic bands as a result of the same charge transfer per alkali atom. Corresponding references have now been added to the revised manuscript on p. 8, second paragraph (refs. 40,41).

Having said that, we also tried our best to understand what caused the referee's disbelief. We realized that our statement about 'exactly the same critical temperature' could be confusing. Of course, one can expect some variations in T_c , even for superconductivity induced by the same degree of electron doping and based on the same intrinsic phonon modes of phosphorene, as suggested in the manuscript.

However, the experiment shows that the difference in the effect of different intercalants is so minor (within 0.2 K) that we could not attribute the detected slight differences in T_c to different intercalating metals. In the original submission, trying to emphasize that all T_c are close, we used – rather unwisely – the wording 'exactly the same' rather than 'the same within the experimental accuracy'. The revised manuscript clarifies this point both in the Abstract and in the main text on p. 4 (first paragraph) and p. 8 (second paragraph).

To this end, we now show our experimental data for different intercalants in more details (new inset in Fig. 2). It is clear that the onset of superconductivity occurs close to 3.8K for all the intercalants but still varies between different compounds (within +/-0.1K as stated previously). Unfortunately, the same variations are also found for different samples using the same intercalant (examples are shown in new Supplementary Fig. S4). These sample-to-sample variations de facto limit our accuracy of determining T_c . The closeness (rather than exactness) of T_c is quite unusual and the main reason why we attribute the superconductivity to 'doped phosphorene' rather than to 'intercalated BP'. The latter would exhibit superconductivity dependent on an intercalant (like in the case of alkali-intercalated graphite compounds) whereas in the former case we expect the transition to be determined mostly by the doping level and phonon modes within phosphorene layers.

Some reasons are listed below:

1. I don't think that the title of "Observation of doping-induced superconductivity in phosphorene" is appropriate to describe the intercalated BP system studied. This can be misleading, because the achieved doped surface (in micron scale) of the black P in this study should be quite different from a doped phosphorene (in nano scale).

The reasoning behind using word phosphorene is explained above. Nonetheless, we agree with the referee that experimentally we deal with intercalated BP and readers should be allowed to make their own conclusions from the observed universality of T_c . The title is now revised to 'Intercalant-independent transition temperature in superconducting black phosphorous'.

2. If the T_c is exactly 3.8 K for all samples, then that should be impossible if they are really doing what they say that they are doing - intercalating the alkali ions. Given that the T_c s are all the same, it seems like that it would be fair for them to have to prove that they really did intercalate the black P to make the superconductor.

3. It can't be excluded that the unchanged T_c are originated from the same thing rather than the doped phosphorene as they expected, for instance, the contamination.

As explained above, we cannot distinguish between different T_c induced by different intercalants. They are just between 3.7 and 3.9 K for all of them. In intercalated MoS_2 , onset T_c is also known to vary within ± 0.1 to ± 0.2 for the same intercalant [e.g. Somoano et al, J. Chem. Phys. 58, 697-701 (1973)].

As concerns possible contamination, let us note that only high-quality reagents (better than 99.9%) were used in the experiments whereas the superconducting volume reached 10%. In addition, the mixture of BP and alkalis did not show any superconducting response until the intercalation reaction was induced. This information is added to the revised manuscript (p.9, Methods). Furthermore, the exposure of the intercalated BP to clean air rapidly extinguished superconductivity as discussed in the manuscript. This unequivocally excludes contamination as a possible source of the observed superconductivity.

In summary, with all our experience in intercalated superconducting compounds, we cannot imagine any other reason for the observed superconductivity than the one discussed in the manuscript. We consider the observed universality of T_c to be highly unusual but not counterintuitive, especially taking into account the similar closeness in T_c observed for MoS_2 with different intercalants and the theory of superconductivity for doped phosphorene. We believe that our experiment firmly points at the latter explanation.

Reviewer #2

In the reply, the authors failed to give convinced explanations for my main concern about the role of the intercalants in developing superconductivity in BP, but they insistently attributed the “unchanged superconducting transition temperatures (T_c)” to the surface layers (which is named as the phosphorene by the authors) based on their results achieved from dc susceptibility measurements. In fact, it is known that the results obtained by this method mainly reflect the bulk nature of the sample, but not the surface property.

In addition, authors used intercalants containing Li, K, Rb, Cs and Ca in their study separately. Actually, the alkalis K, Rb and Cs are commonly used as a series of dopants (while, alkaline earth metal Ca, Sr and Ba belong to another series) to investigate the doping effect on superconductivity for a studied system. Therefore, it is reasonable to compare the doping effects only among K, Rb and Cs, instead of including Li and Ca because of the bigger difference in chemistry between them. The new results (inset of Fig.2a) added by the authors in the revised manuscript show that the T_c s of the samples doped by K, Rb and Cs are clearly increased, which is in conflict with their main conclusion.

Considering the unsatisfied interpretations for the key issue above, I still cannot be convinced to recommend the revised manuscript to be published in NC.

Reply to reviewer #2

In the reply, the authors failed to give convinced explanations for my main concern about the role of the intercalants in developing superconductivity in BP, but they insistently attributed the “unchanged superconducting transition temperatures (T_c)” to the surface layers (which is named as the phosphorene by the authors) based on their results achieved from dc susceptibility measurements. In fact, it is known that the results obtained by this method mainly reflect the bulk nature of the sample, but not the surface property.

We agree that dc magnetization and susceptibility measurements reflect the bulk nature of superconductivity in our samples, as indeed emphasized in the manuscript. We never attributed the superconductivity to ‘surface layers’. Our interpretation is based on doping of *individual layers* of black phosphorous (called phosphorene) by surrounding monolayers of intercalating atoms. Crystallographically speaking, the situation is quite similar to (bulk) intercalated graphite. We brushed through the text again to make sure that readers won’t be confused about surface layers.

In addition, authors used intercalants containing Li, K, Rb, Cs and Ca in their study separately. Actually, the alkalis K, Rb and Cs are commonly used as a series of dopants (while, alkaline earth metal Ca, Sr and Ba belong to another series) to investigate the doping effect on superconductivity for a studied system. Therefore, it is reasonable to compare the doping effects only among K, Rb and Cs, instead of including Li and Ca because of the bigger difference in chemistry between them.

We agree that doping effects from alkaline and alkali-earth metals should be different (after all, they belong to different columns in the periodic table!). As reported in our manuscript, the doping was indeed different: alkali-earth Ca was incorporated into the samples in twice smaller concentrations and, in accordance with literature, per Ca atom transferred twice larger charge to phosphorene layers than the alkali intercalants (resulting in the same overall level of doping as for alkali atoms). We also note in passing that Li belongs to the same chemical group as K, Rb and Cs. Only Ca is an alkali-earth metal in our study.

Despite the referee’s advice, we feel it would be inappropriate to exclude the latter metal from our discussion. Because T_c does not change even for Ca, this makes our observations even more interesting. Let us also mention that, when presented at conferences and workshops, the results generated very significant interest from the community, especially because we were able to compare such a broad range of intercalants.

The new results (inset of Fig.2a) added by the authors in the revised manuscript show that the T_c s of the samples doped by K, Rb and Cs are clearly increased, which is in conflict with their main conclusion.

We thank the referee for looking carefully through our figures and agree that the inset of Fig. 2a seemingly showed a tendency for T_c to slightly ($<0.1K$) increase with increasing the atomic number of alkali metals. However, as emphasized throughout the manuscript, T_c for different samples using exactly the same intercalant varied randomly within a scatter of $0.1K$. The noticed tendency is well within this scatter and, if a different set of data from individual intercalated samples is used, there is no correlation between T_c and the mass/size of the intercalating atoms. The sequence noticed by the referee was just a coincidence, valid for the specific data set. More generally, it would be inappropriate for us to make conclusions from tendencies noticed below experimental accuracy.

To avoid any possible confusion, in the revised manuscript we present data from a different set of individual samples (new inset in Fig. 2a). It shows no correlation between T_c and the mass of the intercalating metal.

Reviewer #1 (Remarks to the Author)

Reviewer #2 opening paragraph: Apart from the restating of the papers work this reviewer doesn't add much apart from their disbelief. I think the rebuttal is very clear and measured. It also provides a context within, and against, which the paper presents the importance of this result. At this point it would be hard to find where this disbelief comes from. In addition they have moderated their paper to address the experimental conditions and consequent analysis. I agree with these changes/modifications. To some extent this addresses aspects of reviewer#2 point 2.

Reviewer #2 Point 1: The authors see the point made and I agree with their change of title to accommodate the reviewers comment.

Reviewer #2 Point 2: One of the main points made and of importance is the fact that T_c remains about 3.8K independent of the mass of the metal cation used as intercalant. This lack of an isotope-like effect points to a different function for the intercalant, namely a charge donor associated with structural modification of the phosphorous sheets. This is the point the authors clearly make.

The authors deal very explicitly with the difficulties raised by the X-ray data, referring to the impact of subjecting the sample to air and the re-emergence of both the BP and CsOH structures. If the Cs had just been sitting on the surface then this would have been present in the X-ray studies. It wasn't in the X-ray pattern. Clearly as I said in my own review this is a discovery paper and the lack of this early definition of crystal structure is not surprising.

Reviewer #2 Point 3: This point is in some way ancillary to point 2 and point to the possibility of a contamination. The authors dealt with this by reference to the purity of the materials they have used. Also they point out that the individual components are not superconductors. I looked at one aspect that has not been covered and that is possibility of the formation of BP-intercalated with metal hydrides. From my knowledge of metal ammonia reactions there is a very small, but viable, chance that ammonia can decompose in the formation of metal ammonia solutions used for graphite intercalation compounds; with the possible formation of small quantities of metal hydrides. The only example of intercalation compounds I could find is from a paper by T. Enoki et al.: Hydrogen-alkali-metal-graphite ternary intercalation compounds (J. Matter. Res. **5**, 435 (1990)). Here Enoki et al looked at C_8K bathed in hydrogen gas (not ammonia liquid).

Compound	Transition temperature (K)
C_8K	0.15
C_8Rb	0.026
$C_8KH_{0.19}$	0.22
$C_8KH_{0.68}$	Does not superconduct down to 0.05

If there were an analogous contaminant the closest from this table above would be $C_8KH_{0.19}$. But the T_c that the authors demonstrate is a factor 10 higher. What is more I would not describe it as a contaminant. I would describe it as one of the physical properties of a newly discovered compound.

Another potential contaminant could be a phosphorus hydride A. P. Drozdov et al: Superconductivity above 100 K in PH_3 at high pressures (condmat). High pressure here means 200 GPa. At ambient pressure PH_3 is a semiconductor. So the superconductivity does not originate from a phosphorus hydride contaminant. In addition, this would show in the X-ray patterns and the compound would not respond to air exposure the way the authors demonstrate experimentally. In short I can see no

clear demonstrable case for a contaminant. As such unless the reviewer #2 has a clear example I see no case for this belief.

In summary I would argue that reviewer #2 has been thin on detail and has ignored a number of aspects which the authors are trying to bring to the attention of the materials community. I believe that authors have addressed reviewer #2 terse comments as best they can. In short in reading this paper again and accepting the modifications I strongly support this paper and can find no clear ground for reviewer #2 opinions.